# CID: Measuring Feature Importance Through Counterfactual Distributions

Eddie Conti[*1], Álvaro Parafita[1], and Axel Brando[1]

[1]TAIES Group, HPES Lab, Barcelona Supercomputing Center (BSC)
{econti,aparafita,abrando}@bsc.es

## Abstract

Assessing the importance of individual features in Machine Learning is critical to understand the model's decision-making process. While numerous methods exist, the lack of a definitive ground truth for comparison highlights the need for alternative, well-founded measures. This paper introduces a novel post-hoc local feature importance method called Counterfactual Importance Distribution (CID). We generate two sets of positive and negative counterfactuals, model their distributions using Kernel Density Estimation, and rank features based on a distributional dissimilarity measure. This measure, grounded in a rigorous mathematical framework, satisfies key properties required to function as a valid metric. We showcase the effectiveness of our method by comparing with well-established local feature importance explainers. Our method not only offers complementary perspectives to existing approaches, but also improves performance on faithfulness metrics (both for comprehensiveness and sufficiency), resulting in more faithful explanations of the system. These results highlight its potential as a valuable tool for model analysis.

## 1 Introduction

Explainable AI (XAI) addresses one of the most pressing challenges in modern Machine Learning: understanding and clarifying how a model produces its outputs. With the extensive adoption of black-box models in sensitive and high-stakes domains—such as medical diagnosis, credit scoring, and criminal justice—the need for methods that are understandable has become not merely desirable but indispensable. In these contexts, the consequences of opaque decision-making can be severe, ranging from unequal treatment to life-altering outcomes ([1], [2], [3]).

One way to tackle this problem is to detect which features are more relevant in generating the output (feature attribution). To do so, the field focused on two classes of methods: intrinsic and post-hoc methods ([4]). Intrinsic methods employ model architectures that are interpretable by design, whereas post-hoc methods operate independently of the model by analyzing its outputs. Which method is preferable is subject of scientific debate, with conflicting opinions and arguments ([5]).

Among post-hoc methods, counterfactual (CF) explanations, first introduced in the context of XAI by [6], have emerged as a popular approach for model interpretability. A CF explanation describes a situation of the form: "If input features X had taken different values, then model's output Y would have been different". Counterfactuals are human-friendly explanations, because they are contrastive to the current instance and because they are often selective, meaning they usually focus on a small number of feature changes ([7, 8]), which highlights the most relevant changes needed to alter the outcome. For instance, if a person named Peter is denied a loan, a counterfactual explanation could state: If Peter earned 10,000 more per year, he would receive the loan.

We propose a novel approach, which we refer to as Counterfactual Importance Distribution (CID), based on the generation of CFs for a specific instance. Specifically, we rank dimensions where positive and negative CFs—i.e., those that succeed and fail in changing the desired outcome, respectively—differ the most. In this way, we are estimating which entries are most important for the model. Section 3 starts with the overview of the proposed framework. In particular, we explain how to use positive and negative CFs and how to model their distributions (Sections 3.1 and 3.2). Then, in Section 3.3, we introduce the intuition and later the definition of formula (2), exploring its properties and geometrical meaning. We conclude the section with two important results, Theorems 3.1 and 3.2, showing that our equation defines a measure of dissimilarity and a (metric) distance. Section 4 is devoted to the experiments; here, we compare CID with DiCE local feature importance scores as well as other metrics on two distinct datasets. In Section 5 we address the limitations of the proposed method as well as possible ways to extend it. In Section 6 we stress on the applicability of our method: it allows for any density estimation and counterfactual generator. We conclude the paper with Section 7 in which we highlight the novelty of CID and the results in terms of faithfulness. In general, the contributions of this work can be summarized as follows:

---

[*]Corresponding Author.

Proceedings of the 7th Northern Lights Deep Learning Conference (NLDL), PMLR 307, 2026.

1. We propose an innovative local feature importance method that uses positive and negative CFs to capture the relevance of features in generating a specific output. Our formulation takes into account support and density of the data, thus capturing nuances in their distribution.

2. We effectively prove that our proposed measure satisfies the mathematical properties required to qualify as a metric distance. This formal grounding ensures that the measure is not only conceptually valid but also mathematically rigorous.

3. We conduct extensive analyses on two datasets, comparing our metric with standard feature importance scores and highlighting its complementary nature. Furthermore, by evaluating local importance scores through comprehensiveness and sufficiency metrics, we demonstrate that our framework consistently yields more faithful and, therefore, reliable results.

## 2 Related Work

Local explainability methods are used to provide insights into individual predictions in Machine Learning models where the relationship between features and outputs is complex or opaque. These methods attempt to explain the model's decision process for a particular prediction, in contrast to global interpretation methods that explain overall model behavior.

Broadly speaking, most local methods focus on calculating the importance of each feature for the specific instance under analysis ([9]). Several methods have been proposed in recent years to tackle this challenge. For instance, Individual Conditional Expectation (ICE), first introduced by [10], is a technique that visualizes how the model's prediction changes when a feature varies for a single instance. LIME (Local Interpretable Model-agnostic Explanations) ([11]) provides another approach, where it approximates the complex model locally with an interpretable surrogate model by perturbing the input data around the instance of interest to fit a simpler model on these perturbed instances. From the fitted local model we obtain explanations for the original model. Despite LIME's popularity, its reliance on local linearity makes it sensitive to the choice of proximity measures and perturbation methods ([12]).

CFs ([6]) offer an alternative by identifying the minimal changes to the input features required to alter a model's prediction. CFs focus on generating "what if" scenarios, offering users a counterfactual instance that illustrates what would need to happen for a different outcome. The main limitation of counterfactual explanations is that they are instance-specific, i.e. no general information about the model

reasoning as a whole is extracted, as shown in [13, 14]. Related to CFs, Anchors [15] describe a prediction as being anchored by certain feature values, which lock the model's prediction in place. Essentially, the method works by finding decision rules that "anchors" the prediction which remains unchanged regardless of variations in other features.

Shapley Values come from cooperative game theory and represent the average marginal contribution of a feature across all possible coalitions. Shapley values assign a score to each feature based on how much it contributes to the prediction in comparison to all other possible subsets of features. SHAP ([16]), an adaptation of Shapley values for machine learning, has become one of the most widely used methods for feature attribution, as it provides a theoretically grounded explanation that satisfies several desirable properties, such as local accuracy, missingness and consistency, even though it may suffer from correlation among features [17].

Although several techniques exist in the literature LIME, SHAP and CFs are among the most popular approaches [18, 19] and have been widely adopted in various applications. However, different feature importance methods can yield different rankings of features, even for the same dataset and model. This is a critical aspect of feature importance methods and stems from many sources as the data complexity, stochastic components of the method or dataset properties ([20], [21], [22]). As highlighted by [5], studies in this field rarely provide a clear justification for choosing one particular method over others. This lack of transparency makes it challenging to evaluate or select the best method for any given problem. Additionally, since there is no universally accepted 'ground truth' in model explainability ([23],[24],[25]), the validity of these methods often relies on empirical validation rather than rigorous proofs. In this context, we compare CID with established methods, leveraging Feature Agreement and faithfulness metrics to provide a fair and comprehensive evaluation. We propose this new method since, as highlighted by studies such as [26] and [5], adopting a plurality of perspectives allows for a more thorough understanding.

Although there are other works that exploit CFs to estimate feature importance ([27],[28], [29]), to the best of our knowledge, our approach of measuring distributional differences stemming from counterfactuals is novel.

## 3 Ranking dimensions with positive and negative CFs

In this section we propose the CID method which consists of three steps: 1) The generation of CFs, presented in Section 3.1. 2) The Kernel Density

Estimation (KDE) modelling (Section 3.2) of the two sets of positive and negative CFs $C^+$ and $C^-$ across each dimension. 3) The rank of numerical features through the $d_1$ distance that we introduce in Section 3.3. Importantly, in our framework the value of $d_1$ for each feature directly represents its feature importance score.

## 3.1 Generating Counterfactuals

Let us consider a black-box classifier $M \colon \mathbb{R}^d \to \{0,1\}$ and a certain input $\tilde{x} \in \mathbb{R}^d$ for which the output $y = M(\tilde{x})$ is observed. Counterfactuals are solutions to:

$$x_{cf} = \arg\min_{x'} \mathcal{L}(M(x'), y_{target}) + \lambda d(\tilde{x}, x')$$

where $\mathcal{L}$ is a loss function, for instance the $L_2$ loss, $\lambda$ is a regularization parameter, $y_{target}$ is the desired outcome and $d(\cdot, \cdot)$ is a distance function ensuring that $\tilde{x}$ and its counterfactual $x'$ remain close. This initial approach, while straightforward, has notable limitations. In particular, it does not penalize unrealistic feature combinations. Building on this foundation, [8], proposed an alternative formulation that minimizes a combined loss function over all generated CFs. In particular, they do not only minimize the distance between the prediction for CFs and the desired outcome, but they also encode proximity, sparsity and diversity, all desirable properties ([7, 30, 31]). The goal is to obtain a variety (diversity) of CFs that that are close to the original input (proximity) and more feasible in the sense of changing fewer number of features (sparsity). In the experiments (Section 4), we employ the DiCE library to create CFs, which make use of this formulation. Such generation often relies on techniques such as trial and error or optimization algorithms like NSGA-II ([32]). The advantages and limitations of CFs have been widely studied in recent years. Please refer to [33] and [34] for a wider discussion on the topic.

## 3.2 Modelling distribution of CFs

Let us assume the generation of two sets of multiple CFs: positive CFs $C^+$ and negative CFs $C^-$, each with cardinality $m$. Positive CFs $C^+$ are those that successfully flip the output $y$, while negative CFs $C^-$ fail to achieve the desired outcome. Our goal is to rank the dimensions in the input space $\mathbb{R}^d$ (or in the activation space of a neural network layer, for neural models) where $C^+$ and $C^-$ differ the most. This ranking provides insights into which dimensions are most critical in determining the model's output.

A straightforward approach is to compute the variability of each dimension between $C^+$ and $C^-$. Specifically, fixed an ordering in $C^+$ and $C^-$, for each dimension $i$ we can compute:

$$\frac{1}{m} \sum_{j=1}^{m} (x_{i,j}^+ - x_{i,j}^-)^2.$$

where $x_{i,j}^+$ and $x_{i,j}^-$ represent the $i$-th entry of the $j$-th counterfactual in $C^+$ and $C^-$ respectively.

While variability provides a simple and effective heuristic, it does not account for the underlying data distribution. To address this, in the following subsection, we introduce a metric function that describes the difference between these distributions; for that reason, we need a way to model the density of sets $C^+$ and $C^-$, for which we employ KDE. The estimated distributions are given by

$$P(x)_i = \frac{1}{mh_1} \sum_{j=1}^{m} K\Big(\frac{x - x_{i,j}^+}{h_1}\Big),$$

$$Q(x)_i = \frac{1}{mh_2} \sum_{j=1}^{m} K\Big(\frac{x - x_{i,j}^-}{h_2}\Big)$$

where $K(\cdot)$ is the kernel function (e.g., Gaussian or Epanechnikov kernel; see [35]), $h_1$ and $h_2$ are bandwidth parameters for $C^+$ and $C^-$. By comparing the above distributions $P(x)_i$ and $Q(x)_i$ for each dimension we can capture more nuanced differences in how positive and negative CFs behave. This method is particularly useful in cases where feature variability alone may not fully capture the importance of a dimension. In order to ground this intuition within a mathematical framework, we introduce a notion that captures how the two distributions differ.

## 3.3 The notion of overlap of functions

To formalize the intuition about how two distributions $p$ and $q$ differ, we introduce a measure of overlap $o(p, q)$. Before presenting the formal definition, we outline some desirable properties of such a measure:

1. If $p = \mathbf{1}_{[0,1]}$[1] and $q = \mathbf{1}_{[1,2]}$, the overlap measure $o(p, q)$ should yield a small value, as the support of $p$ and $q$ intersect in a set of measure zero.

2. On the other hand, if $p = q$ then $o(p, q)$ should attain its maximum value, indicating high overlap.

3. Beyond support overlap, the measure should account for the densities of $p$ and $q$. For instance, if $\text{supp}(p) = [0, 1]$ and $\text{supp}(q) = [0, 100]$, but

$$\int_1^{100} q(x)\, dx \approx \epsilon$$

where $\epsilon$ is a relatively small value, the overlap measure $o(p, q)$ should remain high, indicating a limited contribution of $q$ outside $\text{supp}(p)$.

---

[1]We denote $\mathbf{1}_A$ the indicator function of set $A$.

As a consequence, we define the notion of overlap as

**Definition 1.** Let $p, q$ be real positive integrable functions, the notion of overlap is defined as

$$o(p,q) = \frac{\int_{\text{supp}(p)\cap\text{supp}(q)} \min(p(x),q(x))dx}{\int_{\text{supp}(p)\cup\text{supp}(q)} \max(p(x),q(x))dx} , \quad (1)$$

and given $k \in \mathbb{R}$ such that $k \geq 1$, we define the measure $d_k(p,q)$ as

$$d_k(p,q) = k - o(p,q). \quad (2)$$

Figure 1 provides a graphical example of the computation of the above formula for several variables from the *Diabetes* dataset ([36]). In particular, we use the `np.trapz` method from the `numpy` library, which applies the trapezoidal rule to numerically approximate the value of the integral. Equation (1) is actually a generalization of *Jaccard distance*, a common way to measure dissimilarity between sets.

### 3.3.1 Mathematical and geometrical properties of $d_1$

In this section, we explore a set of properties and results that will help clarify the meaning of the overlap measure and provide the foundation for the main theoretical results of this paper: Theorems 3.1 and 3.2. The discussion will be done for $k = 1$ as it is the most relevant; however, all the results can be immediately adapted to $d_k(p,q)$. First of all, note that since $p(x), q(x) \geq 0$ and

$$\text{supp}(p) \cap \text{supp}(q) \subset \text{supp}(p) \cup \text{supp}(q),$$

we can conclude that

$$0 \leq o(p,q) \leq 1$$

which implies that

$$0 \leq d_1(p,q) \leq 1.$$

If $p = q$ then $d_1(p,q) = 0$ and if the two density estimations share a 0-measure support i.e., $\mu(\text{supp}(p) \cap \text{supp}(q)) = 0$ for a given measure $\mu$, we obtain that $d_1(p,q) = 1$. Furthermore, we have a monotonicity property that takes into account densities: if $p(x), g(x), q(x)$ are three density estimations sharing the same support with the property that, almost everywhere,

$$p(x) \leq g(x) \leq q(x) \implies d_1(p,q) \geq d_1(g,q).$$

It is important to underline that, since we are working with integrals, we have to consider statements as $p < q$ or $p \neq q$ almost everywhere. More properly, equations (1) and (2) are defined on

$$L^1(\mathbb{R}) \times L^1(\mathbb{R}) = \frac{\mathcal{L}^1(\mathbb{R}) \times \mathcal{L}^1(\mathbb{R})}{\sim_\mu}$$

which is the quotient space of $\mathcal{L}^1(\mathbb{R}) \times \mathcal{L}^1(\mathbb{R})$ with the equivalence relation induced by $\mu$. Finally, we are left to show that $d_1(p,q)$ satisfies property 3 in Section 3.3.

**Proposition 3.1.** *Let $p, q$ be two real probability distributions, $supp(p) = [a,b]$, $supp(q) = [a,c]$ such that $b < c$. Moreover, assume that*

$$|p(x) - q(x)| < \delta \quad \text{for } x \in [a,b], \qquad \int_b^c q(x) = \epsilon.$$

*Then $o(p,q) \to 1$, and so $d_1(p,q) \to 0$, as $\epsilon, \delta \to 0$.*

*Proof.* We leave the proof for Appendix A.1. $\square$

Several density functions are defined on $\mathbb{R}$, even though they are concentrated on a small region as in the case of $N(\sigma, \mu^2)$ and $\Gamma(\alpha, \beta)$ (a list of Kernel functions can be found in [35]). Assume now that $p(x), q(x)$ are two densities defined on $\mathbb{R}$, then we can concentrate our overlap measure on a finite proper interval $D$ without any relevant information loss. This is a result of the continuity of $d_1(p,q)$, Proposition 3.1 and the following remark, for which we leave additional details in the Appendix A.2:

*Remark* 3.1. Let $f \in L^1(\mathbb{R})$, then $\forall \epsilon > 0, \exists a \geq 0$ such that

$$\int_a^\infty |f(x)|\, dx < \epsilon, \quad \int_{-\infty}^{-a} |f(x)|\, dx < \epsilon.$$

Let us now provide a numerical example of computation of $d_1(p,q)$.

*Example* 3.1. Assume the following density functions

$$p(x) = \begin{cases} \frac{1}{2} & \text{if } x \in [0,2] \\ 0 & \text{otherwise} \end{cases}, \quad q(x) = \begin{cases} \frac{19}{40} & \text{if } x \in [0,2] \\ \frac{1}{60} & \text{if } x \in (2,5] \\ 0 & \text{otherwise.} \end{cases}$$

In this case, the overlap value is $19/20$ and so

$$d_1(p,q) = 1 - \frac{19/20}{1 + 1/20} = \frac{2}{21} \approx 0.095,$$

indicating little dissimilarity between the two densities.

We prove the following Proposition and Theorem in Appendix A.3.

**Proposition 3.2.** *Given $p, q$ real positive integrable functions such that $p \neq q$, then $d_k(p,q) > k - 1$.*

**Theorem 3.1.** *The measure $d_k(p,q)$ is a metric dissimilarity[2] measure on $\Omega$ for $k \geq 2$, where*

$$\Omega = \{f \in L^1(\mathbb{R}) \mid supp(f) \neq \emptyset \text{ and } f(x) \geq 0\}.$$

---

[2]The formal definition of dissimilarity can be found in the Appendix. Essentially, it is related to the concept of metric on a set, but with weaker requirements.

Our focus is the case $k = 1$, for which we can obtain a stronger result for $d_1(p, q)$. In short, invoking previous results, using the fact that the *Jaccard distance* satisfies the triangle inequality and by dominated convergence, we can obtain (see Appendix A.4) the following central Theorem, which also generalize the previous result:

**Theorem 3.2.** $d_1(p, q)$ *is a metric on* $\Omega$. *Moreover,* $d_k(p, q)$ *is a dissimilarity measure on* $\Omega$ *for* $k \in [1, 2)$.

As a consequence, Theorems 3.1 and 3.2 convey that formula defined in (2) represents a way of calculating in mathematical terms the dissimilarity between two probability distributions. Moreover, Theorem 3.2 establishes that $d_1(p, q)$ satisfies the axioms of a distance: non-negativity, symmetry, the triangle inequality, and that $d_1(p, q) = 0$ if and only if $p = q$ (almost everywhere). This result is significant because it implies that $d_1$ induces a metric topology on the space $\Omega$. This topology provides a rigorous framework for comparing probability distributions. From a topological perspective, the metric $d_1$ ensures that the space $\Omega$ is metrizable. For instance, $d_1$ can be used to define neighborhoods of probability distributions, facilitating the study of their stability, convergence, or variation under perturbations.

# 4 Experimental results

To demonstrate the applicability of the proposed framework, we conduct experiments on numerical features from two datasets: *Diabetes* and *Heart Disease* ([36],[37]). To showcase the effectiveness of CID, we deliberately adopted standard techniques (alternative settings are discussed in the Appendix A.5): Gaussian Kernel with Silverman's rule for the bandwidth to compute the (approximate) distributions of $C^+$ and $C^-$; on the other hand, counterfactual explanations are computed using the Diverse Counterfactual Explanations (DiCE) library ([8]), which some authors regard as a benchmark for evaluating feature importance ([28, 38]). We employ this library as it allows to generate positive and negative CFs for any model. We briefly recall that a generated point is classified as a positive CF if it crosses the decision boundary and changes the classifier output; otherwise, it is labeled as a negative CF. In the experiments we generate 50 elements for each set $C^+$ and $C^-$. We take inspiration from [39] and compare our results (CID) with local feature importance scores from DiCE. In order to provide a complete picture, we also compute SHAP values ([16]) and LIME values ([11]) for the analyzed variables.

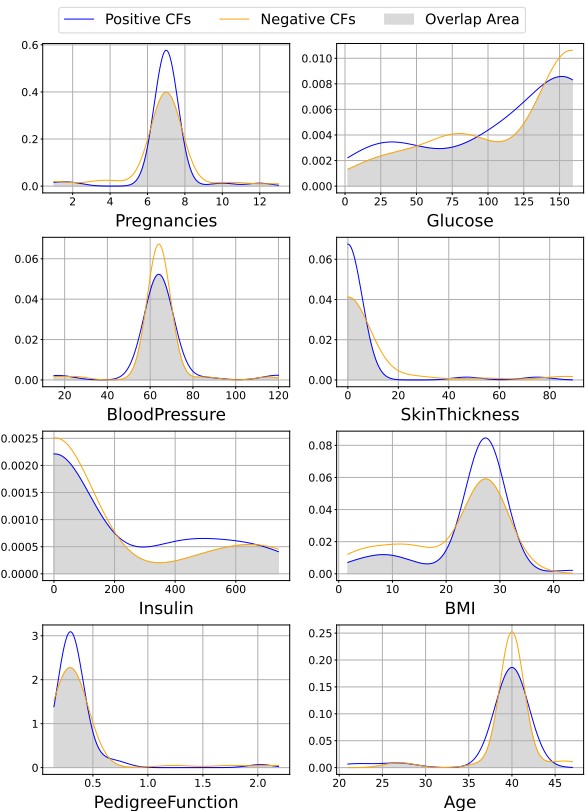

**Figure 1.** Dissimilarity analysis for the *Diabetes* dataset. KDEs of CF entries from $C^+$ (blue) and $C^-$ (orange) for the first test sample; the grey area indicates their overlap. This visualization highlights how feature distributions differ between positive and negative CFs. For instance, SkinThickness and PedigreeFunction show noticeable shifts, while variables like Insulin and Glucose exhibit no clear pattern.

## 4.1 Dissimilarity analysis on datasets

Figure 1 shows a graphical illustration for the *Diabetes* dataset of the computation of $d_1(p, q)$ from the two sets of CFs (positive and negative). We recall that $d_1(p, q)$ conveys how different two distributions are: a higher $d_1(p, q)$ indicates that the corresponding feature assumes substantially different values in positive and negative counterfactuals, meaning that changes in this feature are more strongly associated with changes in the model output. In addition to provide an illustrative comparison between CID and DiCE, both relying on generation CFs, we computed the value of CID and DiCE for the first entry (for reproducibility) of *Diabetes* and *Heart* datasets. In Figure 2 we compare the distributions of the obtained values, replicated 100 times, for each feature. In the case of CID, we can see a higher variability compared to DiCE, reflecting the variance from the CFs generation process, a know issue in feature importance methods [20, 21, 40, 41].

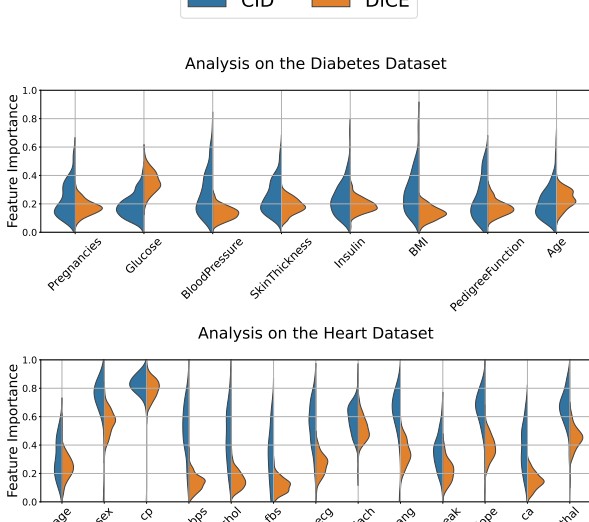

**Figure 2.** Distribution of feature importance values over 100 iterations, according to CID and DiCE, for the first test entry of the *Diabetes* and *Heart* datasets, respectively. We employed, respectively, LogisticRegression and RandomForestClassifier as models.

**Table 1.** Feature Agreement Matrix for the *Diabetes* dataset. Values highlight the lower alignment of CID with the other metrics, hence suggesting different dimensions as relevant.

|  | DiCE | SHAP | LIME | CID (ours) |
|---|---|---|---|---|
| DiCE | $1.00 \pm 0.00$ | $0.67 \pm 0.03$ | $0.71 \pm 0.03$ | $0.46 \pm 0.04$ |
| SHAP | $0.67 \pm 0.03$ | $1.00 \pm 0.00$ | $0.84 \pm 0.02$ | $0.57 \pm 0.03$ |
| LIME | $0.71 \pm 0.03$ | $0.84 \pm 0.02$ | $1.00 \pm 0.00$ | $0.53 \pm 0.03$ |
| CID (ours) | $0.46 \pm 0.04$ | $0.57 \pm 0.03$ | $0.53 \pm 0.03$ | $1.00 \pm 0.00$ |

**Table 2.** Feature Agreement Matrix for the *Heart* dataset. In this table, CID shows an even more pronounced misalignment in the features suggested as relevant compared to the other metrics.

|  | DiCE | SHAP | LIME | CID (ours) |
|---|---|---|---|---|
| DiCE | $1.00 \pm 0.00$ | $0.53 \pm 0.03$ | $0.71 \pm 0.02$ | $0.35 \pm 0.05$ |
| SHAP | $0.53 \pm 0.03$ | $1.00 \pm 0.00$ | $0.69 \pm 0.03$ | $0.33 \pm 0.03$ |
| LIME | $0.71 \pm 0.02$ | $0.69 \pm 0.03$ | $1.00 \pm 0.00$ | $0.35 \pm 0.04$ |
| CID (ours) | $0.35 \pm 0.05$ | $0.33 \pm 0.03$ | $0.35 \pm 0.04$ | $1.00 \pm 0.00$ |

## 4.2 Comparison with other measures

In this section, we compare CID with DiCE, SHAP and LIME. Moreover, to mitigate the randomness of CFs, we repeat the analysis 10 times, averaging the results to compute the dissimilarity measure $d_1(p,q)$ and the DiCE local feature importance. Next, we aggregate the results across the entire test set for each dataset to ensure a complete comparison across the metrics. In the case of *Diabetes* we use LogisticRegression, whereas for Heart Disease we use RandomForestClassifier. It is important to emphasize that the same analyses could be conducted with other types of classifiers, as all the measures we use are model-agnostic.

To compare these methods, we use *Feature Agreement* ([20]) with $k = 4$ to evaluate the agreement between methods when selecting the top-k features according to their attribution values (here $k$ refers to the features, and no confusion should arise with $k$ in (2) since we are using $d_1$). More precisely, given two explanations (i.e., vectors consisting of feature importance values) $E_a$ and $E_b$, Feature Agreement is formulated as:

$$\frac{|TF(E_a, k) \cap TF(E_b, k)|}{k} \tag{3}$$

where $TF(E, k)$ returns the set of top-k features of an explanation $E$ based on the magnitude of their feature importance values. Formula (3) computes the fraction of common features between the sets of top-k features of two explanations.

The results in Tables 1 and 2 show the aggregation of (3) for the whole test set. We report the mean

with the confidence interval, computed as $\pm 2\sigma/\sqrt{n}$. From the tables, we can observe how CID is in lower agreement with the other metrics, which in turn are more aligned with themselves. This aspect is even more pronounced in the *Heart* dataset. As a consequence, considering the top features, CID provides *complementary* explanations of the model (focusing on different dimensions of the data), while DiCE, SHAP and LIME tend to highlight the same features with high consistency. This observed discrepancy aligns with the literature on feature importance ([25], [5], [42]), which highlights the lack of ground truth in feature importance attribution. This raises the need to evaluate and quantify the quality of explanations provided by these techniques. In the following section we address this aspect by employing faithfulness metrics.

## 4.3 Evaluation of Faithfulness metrics

The results obtained by using the Feature Agreement measure (3) help understand whether there are differences in stating which features are more important. To assess the explanatory value of these methods, We introduce faithfulness metrics, namely *sufficiency* and *comprehensiveness*, initially proposed by [43] and further discussed in [44] and [45]. These metrics evaluate how well the identified features contribute to the model's decision process. These metrics provide a robust framework for understanding the faithfulness of explanations. As we show in this section, CID achieves significantly better results in terms of faithfulness in our experiments compared to the other approaches. The key intuition behind these metrics is that altering an important feature should significantly impact the model's prediction and the magnitude of this impact reflects the quality of the explanation.

Let us now formalize *comprehensiveness* and *sufficiency* metrics. Given an input $x = (x_1, \ldots, x_d)$ and an explanation $e = (e_1, \ldots, e_d)$ where $d$ denotes the number of features, if we denote $\tilde{x}_e^{(l)}$ the input with $l$ most important features *removed* according to $e$, comprehensiveness is defined as

$$k(x, e) = \frac{1}{d+1} \sum_{l=0}^{d} f(x) - f(\tilde{x}_e^{(l)}), \qquad (4)$$

where $f$ is the function we want to explain. In simple terms, comprehensiveness measures how much the model prediction deviates from its original value when important features are removed sequentially (larger values indicate better explanations). If we denote $\hat{x}_e^{(l)}$ the input with the $l$ most important features present, sufficiency is defined as

$$\sigma(x, e) = \frac{1}{d+1} \sum_{l=0}^{d} f(x) - f(\hat{x}_e^{(l)}) \qquad (5)$$

and it measures the gap to the original model prediction that remains when features are successively inserted from the most important to the least. Here, a smaller value is desirable.

In our experiments, we compute (4) and (5) using

$$f(x) = P(y = M(x)|x),$$

where $M(x)$ represents the predicted target for the model for the particular instance $x$, again reporting the mean value and the confidence interval as done in Section 4.2. The explanations are generated using CID, DiCE, SHAP and LIME. In order to obtain $\tilde{x}_e^{(l)}$ and $\hat{x}_e^{(l)}$, where we *remove* information, we mask selected features by replacing them with the mean value of that dimension ([46]). As an example, Figure 3 shows the evolution of the predicted probability when we progressively mask the input according to the explainers used. This figure is intended as an illustrative example to show the general trend of comprehensiveness. Non-monotonic variations in the predicted probability may arise from feature interactions, correlations, or model non-linearities. The quantitative and statistically supported results are presented in the tabels below.

**Table 3.** Comprehensiveness results for *Diabetes* and *Heart* datasets (error bars with $\pm 2\sigma/\sqrt{n}$). Higher values are preferable.

| Method | Diabetes ↑ | Heart ↑ |
|---|---|---|
| CID | **0.5405 ± 0.2559** | **1.8017 ± 0.0965** |
| DiCE | 0.0487 ± 0.1250 | 1.2355 ± 0.0849 |
| SHAP | 0.1300 ± 0.1423 | 1.3689 ± 0.0716 |
| LIME | 0.1277 ± 0.1328 | **1.8163 ± 0.0815** |

In terms of comprehensiveness in Table 3, CID significantly outperforms the other three methods

**Table 4.** Sufficiency results for *Diabetes* and *Heart* datasets (error bars with $\pm 2\sigma/\sqrt{n}$). Smaller values are preferable.

| Method | Diabetes ↓ | Heart ↓ |
|---|---|---|
| CID | **−0.1288 ± 0.1444** | **2.2129 ± 0.1357** |
| DiCE | 0.3942 ± 0.2758 | 2.7227 ± 0.1166 |
| SHAP | 0.3748 ± 0.2797 | 2.7044 ± 0.1307 |
| LIME | 0.3699 ± 0.2799 | **2.2662 ± 0.1346** |

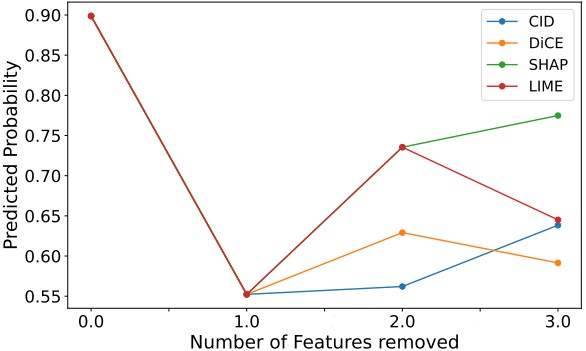

**Figure 3.** The trend of $k(x, e)$ when gradually removing the relevant features according to the different explanations in *Diabetes* dataset for an instance of the test set. A downward trend indicates that features are being removed that actually played a relevant role in determining the class for the instance under consideration.

in the *Diabetes* dataset, while both CID and LIME achieve the best results in the *Heart* dataset. A similar situation can be seen for the sufficiency metrics (here smaller values are preferable) in Table 4, where CID achieves the best results in *Diabetes* and shares the best performance with LIME on the *Heart* dataset.

Overall, even though LIME also demonstrates competitive performance, CID stands out as the more faithful method across both datasets.

# 5 Limitations and future work

The formula (2) provides a method to quantify the dissimilarity between two data distributions, offering a novel perspective on how features contribute to a given output. This work represents an initial exploration of this dissimilarity framework, showing promising results in terms of faithfulness. Importantly, this approach can naturally extend beyond binary classifiers. For instance, in the case of a continuous output space $\mathbb{R}$, it is sufficient to define a subset $A \in \mathbb{R}$ (this could be an interval around the factual sample) where $C^+$ represents the data points for which $f(x) \in A$, and $C^-$ corresponds to those with $f(x) \in A^c$. In our analysis of feature importance using this framework, we assumed feature independence, a common simplification in mathematical modeling [9, 47]. However, this assumption

stems from the generation process adopted by DiCE with random sampling, which treats features independently by design. Importantly, our method is general: if the generation of $C^+$ and $C^-$ were to incorporate feature dependencies, then the marginal distributions analyzed by our metric $d_k$ would naturally capture such correlations. Lastly, the masking method we adopted in the evaluation of faithfulness may lead to biased estimations.

Several open questions remain. Future studies could explore the impact of dependencies among features or extend this framework to multivariate settings and categorical variables. While one-hot encoding would technically enable the application of our method, KDE struggles with the limited variability of categorical features, which leads to volatile and less representative analyses. Moreover, CID is dependent on the CF generation mechanism of DiCE and the KDE estimations; this represents a promising direction for future work, as our framework will benefit from improved CF generation methods and more accurate distribution estimations.

# 6 Applicability of CID and Computational Cost

In Section 4, we demonstrated the effectiveness of CID, which achieves competitive results in terms of both sufficiency and comprehensiveness. To emphasize the robustness of our approach, we deliberately adopted standard techniques: Gaussian kernel density estimation using Silverman's rule and DiCE with random sampling for generating counterfactuals. This was done to highlight the strength of our framework even under simple, baseline conditions.

It is important to underline, however, the *flexibility* of our method. At the core of CID lies the $d_k$ metric, which serves as a bridge between the $C^+$ and $C^-$ distributions and the final feature importance scores. Therefore, the framework is modular: given any density estimation technique and a generative counterfactual method (capable of producing negative counterfactuals), CID can compute feature importance scores via $d_k$. In the Appendix A.5, we conduct experiments on both datasets using various kernels and counterfactual generation methods. Overall, the results show that the standard setting offers a balanced trade-off: better comprehensiveness, comparable sufficiency, and lower computational cost than alternative counterfactual generation methods.

To analyze the computational cost, let $D$ be the dimensionality of the dataset and $m$ the size of each counterfactual set $C^+$ and $C^-$. For a given input $x$, the method generates two counterfactual sets of size $m$ each, then estimates the densities $p_i$ and $q_i$ for each feature $i$ independently (e.g., using a Gaussian kernel density estimator). Finally, the distance $d_1$ between the two densities is computed by numerically integrating two terms via the trapezoidal rule with $n_{\mathrm{grid}}$ discretization points. The integration step has cost $O(n_{\mathrm{grid}}D)$ once the densities are available. The overall complexity can be expressed as three distinct operations—counterfactual generation ($2m$ points), density estimation ($D$ dimensions), $d_1$ computation ($O(n_{grid}D)$). From our empirical observations, the CF generation tends to dominate the runtime: changing the counterfactual generation method noticeably increased the computation time, while switching between different kernel estimators had negligible impact. Therefore, the framework can be optimized depending on the choice of hyperparameters, and the computation of $d_1$ does not appear to be a major bottleneck.

# 7 Conclusions

In the field of model explainability, there is no single definitive approach that prevails universally. Adopting a plurality of perspectives allows for a more comprehensive understanding of the model's decision process. In the case of the $d_1$ metric introduced in (2), Section 4.2 shows that CID captures complementary aspects to other well-established methods, which tend to agree more strongly with each other. Furthermore, when evaluating faithfulness, a critical measure of explanation correctness, Section 4.3 shows that the CID framework in standard setting generally outperforms the other methods in terms of comprehensiveness and sufficiency. Consequently, we believe that the method we have introduced represents a promising starting point for the study of local feature importance. It is well-founded in mathematical terms and offers valuable insights. Further research, such as extending it to multivariate settings, exploring other domains (image, text), will enhance the applicability of the $d_1$ metric and our framework.

# Acknowledgments

The research leading to these results has been supported by the Horizon Europe Programme under the AI4DEBUNK Project (https://www.ai4debunk.eu), grant agreement num. 101135757. Also, by project PLEC2023-010240 funded by MICIU/AEI/10.13039/501100011033 (CAPSUL-IA), and project PID2024-155745OB-I00 funded by Spanish MICIU/AEI/10.13039/501100011033 / FEDER, UE (IAFE). It has also been partially supported by the predoctoral grants FI-STEP (2025 STEP 00108) from the Research and University Department of the Generalitat de Catalunya co-funded by the "Fondo Social Europeo Plus", and JDC2022-050313-I funded by MCIN/AEI/10.13039/501100011033al by Euro-

pean Union NextGenerationEU/PRTR, and the "Generación D" initiative, Red.es, Ministerio para la Transformación Digital y de la Función Pública, for talent attraction (C005/24-ED CV1), funded by the European Union NextGenerationEU funds, through PRTR.

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

# A  Appendix

## A.1  Proof of Proposition 3.1

**Proposition** *Let $p, q$ be two real probability distributions, $supp(p) = [a, b]$, $supp(q) = [a, c]$ such that $b < c$. Moreover, assume that*

$$|p(x) - q(x)| < \delta \quad for\ x \in [a, b], \qquad \int_b^c q(x) = \epsilon.$$

*Then $o(p, q) \to 1$, and so $d_1(p, q) \to 0$, as $\epsilon, \delta \to 0$.*

*Proof.* We prove this Proposition for the general $d_k(p, q)$, then it sufficient to set $k = 1$ at the end. Under these assumptions,

$$\begin{aligned}
d_k(p, q) &= k - \frac{\int_a^b \min(p(x), q(x))dx}{\int_a^c \max(p(x), q(x))dx} \\
&= k - \frac{\int_a^b \min(p(x), q(x))dx}{\int_a^b \max(p(x), q(x))dx + \int_b^c q(x)dx} \\
&= k - \frac{\int_a^b \min(p(x), q(x))dx}{\int_a^b \max(p(x), q(x))dx + \epsilon}.
\end{aligned}$$

Now, from the last equation, we can obtain:

$$k - \frac{1 - \epsilon}{1 - \delta(b - a)} \le d_k(p, q) \le k - \frac{1 - \epsilon - \delta(b - a)}{1 + \delta(b - a)},$$

since

$$\int_a^b q(x) = 1 - \epsilon.$$

The inequality can be obtained by observing that $|p(x) - q(x)| < \delta$ and so $q(x) - \delta < p(x) < q(x) + \delta$ which implies:

$$\min(p(x), q(x)) \le q(x), \quad \max(p(x), q(x)) \ge q(x) - \delta$$

and

$$\begin{aligned}
\min(p(x), q(x)) &\ge q(x) - \delta, \\
\max(p(x), q(x)) &\le q(x) + \delta.
\end{aligned}$$

Now, as desired, if $\epsilon$ and $\delta$ are relatively small, $d_k(p, q)$ takes on a value close to $k - 1$ and so $o(p, q) \to 1$. $\qquad \square$

## A.2 Details on Remark 3.1

**Remark** *Let $f \in L^1(\mathbb{R})$, then $\forall \epsilon > 0$, $\exists a \in \mathbb{R}_{\geq 0}$ such that:*

$$\int_a^\infty |f(x)| \, dx < \epsilon, \quad \int_{-\infty}^{-a} |f(x)| \, dx < \epsilon.$$

In order to demonstrate the statement in Remark 3.1, we observe that since $f \in L^1$ then, by definition,

$$\int_{-\infty}^{+\infty} |f(x)| \, dx = M \geq 0.$$

By continuity of the integral function, as $a \to \infty$,

$$\int_{-a}^{+a} |f(x)| \, dx \to \int_{-\infty}^{+\infty} |f(x)| \, dx.$$

As a consequence, it cannot be the case that exists $\epsilon > 0$ such that for any $a \geq 0$,

$$\int_a^{+\infty} |f(x)| \, dx > \epsilon \quad \text{or} \quad \int_{-\infty}^{-a} |f(x)| \, dx > \epsilon.$$

## A.3 Derivation of Theorem 3.1

**Proposition** *Given $p, q$ real positive integrable functions such that $p \neq q$ (almost everywhere), then $d_k(p, q) > k - 1$.*

*Proof.* Consider two functions $p, q$ such that $p \neq q$. It is straightforward that if $\text{supp}(p) \neq \text{supp}(q)$ then $d_k(p, q) > k - 1$ by how the measure of overlap is defined. Indeed, in the worst case scenario $p = q$ in $\text{supp}(p) \cap \text{supp}(q)$, but with $\text{supp}(p) \neq \text{supp}(q)$, then:

$$\text{supp}(p) \cap \text{supp}(q) \subsetneq \text{supp}(p) \cup \text{supp}(q)$$

which implies

$$o(p, q) = \frac{\int_{\text{supp}(p) \cap \text{supp}(q)} \min(p(x), q(x)) dx}{\int_{\text{supp}(p) \cup \text{supp}(q)} \max(p(x), q(x)) dx} < 1.$$

As a consequence, let us assume $\text{supp}(p) = \text{supp}(q) = D$. Since $p \neq q$, at least one among

$$A = \{x \in D \mid p(x) < q(x)\},$$
$$B = \{x \in D \mid q(x) < p(x)\}$$

has a positive measure. Assume $\mu(A) > 0$, therefore,

$$\frac{\int_D \min(p(x), q(x)) dx}{\int_D \max(p(x), q(x)) dx} = \frac{\int_A p(x) + \int_{D \setminus A} q(x)}{\int_A q(x) + \int_{D \setminus A} p(x)} < 1$$

because

$$\int_{D \setminus A} q(x) \leq \int_{D \setminus A} p(x),$$

$$\int_A p(x) < \int_A q(x).$$

The case $\mu(B) > 0$ is analogous. $\qquad\square$

Now let us recall the following definition.

**Definition 2.** A dissimilarity measure (DM) $d$ on $X$ is a function $d \colon X \times X \to \mathbb{R}$ such that

$$\exists d_0 \text{ s.t. } -\infty < d_0 \leq d(x, y) < \infty \quad \forall x, y \in X,$$
$$d(x, x) = d_0 \quad \forall x \in X,$$
$$d(x, y) = d(y, x) \quad \forall x, y \in X.$$

If in addition

$$d(x, y) = d_0 \iff x = y,$$
$$d(x, z) \leq d(x, y) + d(y, z) \quad \forall x, y, z \in X,$$

$d$ is called a metric DM on $X$.

Let us investigate the triangular inequality,

$$d_k(p, r) \leq d_k(p, q) + d_k(q, r)$$

which in our case becomes:

$$k - o(p, r) \leq k - o(p, q) + k - o(q, r).$$

Simplified, it becomes:

$$o(p, q) + o(q, r) - o(p, r) \leq k$$

which is always satisfied if $k \geq 2$ because every term on the left is positive and bounded by 1. As a consequence of this result combined with Proposition 3.2, noticing that the symmetry is guaranteed and that $d_0$ in Definition 2 in our case is $k - 1$, we have the following theorem:

**Theorem A.1.** *The measure $d_k(p, q)$ is a metric dissimilarity measure on $\Omega$ for $k \geq 2$, where*

$$\Omega = \{f \in L^1(\mathbb{R}) \mid supp(f) \neq \emptyset \text{ and } f(x) \geq 0\}.$$

## A.4 Derivation of Theorem 3.2

**Theorem** *$d_1(p, q)$ is a metric on $\Omega$. Moreover, $d_k(p, q)$ is a dissimilarity measure on $\Omega$ for $k \in [1, 2)$.*

According to our definition, we are left to analyze the case for $k \in [1, 2)$ in (2). First of all, note that the Jaccard distance defined for two sets $A, B$:

$$d_J(A, B) = 1 - \frac{|A \cap B|}{|A \cup B|}$$

satisfies the triangle inequality (cfr. [48]). As a consequence, if we consider the functions $p(x) = \mathbf{1}_A, q(x) = \mathbf{1}_B, r(x) = \mathbf{1}_C$,

$$d_1(p, r) \leq d_1(p, q) + d_1(q, r)$$

and of course if we add on the lhs $k - 1$ and on the rhs $2(k - 1)$ the inequality still holds, hence:

$$d_k(p, r) \leq d_k(p, q) + d_k(q, r).$$

Now, the idea is to prove if the inequality is valid for general characteristic functions. Therefore, if:

$$d_k(\alpha \mathbf{1}_A, \gamma \mathbf{1}_C) \leq d_k(\alpha \mathbf{1}_A, \beta \mathbf{1}_B) + d_k(\beta \mathbf{1}_B, \gamma \mathbf{1}_C)$$

for $\alpha, \beta, \gamma \in \mathbb{R}_{\geq 0}$. Without loss of generality (as we will se later), we can assume that $\alpha \leq \beta \leq \gamma$, so we have to prove that:

$$1 - \frac{\alpha|A \cap C|}{\gamma|A \cup C|} \leq 1 - \frac{\alpha|A \cap B|}{\beta|A \cup B|} + 1 - \frac{\beta|B \cap C|}{\gamma|B \cup C|},$$

which can be rewritten as:

$$\frac{\beta|B \cap C|}{\gamma|B \cup C|} + \frac{\alpha|A \cap B|}{\beta|A \cup B|} - \frac{\alpha|A \cap C|}{\gamma|A \cup C|} \leq 1$$

but this is true, since we know that:

$$\frac{|B \cap C|}{|B \cup C|} + \frac{|A \cap B|}{|A \cup B|} - \frac{|A \cap C|}{|A \cup C|} \leq 1,$$

$$\text{because } 0 \leq \frac{\alpha}{\beta}, \frac{\alpha}{\gamma}, \frac{\beta}{\gamma} \leq 1.$$

We observe that the order of $\alpha, \beta, \gamma$ is irrelevant since we have a minimum over a maximum and so the ratio is always less or equal than 1. Now, let us assume that $p(x), q(x), r(x)$ are simple functions, i.e.,

$$p(x) = \sum_{i=1}^{s} a_i \mathbf{1}_{A_i}, \quad q(x) = \sum_{i=1}^{s} b_i \mathbf{1}_{B_i},$$

$$r(x) = \sum_{i=1}^{s} c_i \mathbf{1}_{C_i},$$

where $s \in \mathbb{N}$, $a_i, b_i, c_i \geq 0$ and $A_i, B_i, C_i$ are disjoint sets. It is trivial now, since the inequality holds for general characteristic functions, that:

$$d_k(p, r) \leq d_k(p, q) + d_k(q, r).$$

Any function in $L^1(\mathbb{R})$ can be approximated with a sequence of simple functions (cfr. [49]). By the dominated convergence theorem, we can exchange the limit with the integral and so, combining it with the results for simple functions, we conclude that:

$$d_k(p, r) \leq d_k(p, q) + d_k(q, r) \quad p, q, r \in L^1(\mathbb{R}).$$

As a conclusion, we can extend Theorem 3.1 to the case $k \geq 1$, and so we obtain Theorem 3.2.

## A.5 Hyperparameters of CID

As discussed in section 6, CID's flexibility enables the use of any density estimator and counterfactual generator. In this section, we showcase the applicability of our framework evaluating combinations of three kernel density estimators—Gaussian, Epanechnikov, and Exponential—with random counterfactual generators. Additionally, we assess the effect of

changing the counterfactual generation method to 'genetic' [50]. The 'gradient' method was not applicable in our setup due to the use of a non-differentiable model (RandomForestClassifier). The comparison is carried out at three levels for 100 instances:

- Feature Agreement: we compute the top-k with $k = 4$ between feature importance vectors across methods.

- Distributional Analysis: we visualize the distribution of importance scores per method.

- Faithfulness: we compute sufficiency and comprehensiveness to compare methods.

The experiments reveal a general alignment between the Epanechnikov and exponential kernels (we note that changing the kernel had no effect in terms of computational time), as evidenced by both the feature agreement and the distribution plots, but they diverge noticeably from those obtained using the Gaussian kernel (Table A.1, Table A.2, Figure A.1 and Figure A.2). This pattern is consistent across both datasets. A possible explanation is that Epanechnikov and exponential kernels both decay more rapidly than the Gaussian kernel. This leads to sharper density estimations, which in turn results in more aligned feature importance profiles. The Gaussian kernel, being smoother and more globally sensitive, captures broader structures and therefore introduces differences both in the density estimation and in the resulting scores.

Interestingly, the choice of counterfactual generation method appears to have a limited impact on the overall distributional results (Figure A.3 and Figure A.4)—despite the fact that the genetic method required more than five times the computation time compared to the random baseline. However, when analyzing feature agreement ($0.16 \pm 0.03$ for the Diabetes dataset and $0.16 \pm 0.04$ for the Heart dataset w.r.t the baseline setting), which is sensitive to even small shifts in local feature importance, the choice of counterfactual generation method plays a more nuanced role.

To conclude the analysis, we report the faithfulness results for the diabetes dataset under the considered settings (Table A.3 and Table A.4). Gaussian kernel is significantly better than the alternatives in terms of comprehensiveness, but there are no conclusive results for any kernel in terms of sufficiency. Overall, despite increased variance in both faithfulness metrics, the Gaussian kernel achieves better results, also considering that changing the CF method had a significant impact in computational terms.

In conclusion, the results of this section indicate that both the choice of kernel density estimator and the counterfactual generation method influence the

resulting attribution method. However, the substantially higher computational cost of the genetic approach makes its adoption generally impractical. From a faithfulness perspective, the standard setting—Gaussian kernel combined with randomly generated counterfactuals—achieves a markedly higher comprehensiveness. When these findings are considered alongside the other results presented in this work (subsection 4.3), they suggest that the standard configuration represents a sound choice.

**Table A.1.** Feature agreement values across different kernels for the Diabetes dataset.

|  | Gaussian | Epanechnikov | Exponential |
|---|---|---|---|
| Gaussian | $1.00 \pm 0.00$ | $0.24 \pm 0.05$ | $0.27 \pm 0.05$ |
| Epanechnikov | $0.24 \pm 0.05$ | $1.00 \pm 0.00$ | $0.60 \pm 0.06$ |
| Exponential | $0.27 \pm 0.05$ | $0.60 \pm 0.06$ | $1.00 \pm 0.00$ |

**Table A.2.** Feature agreement values across different explainers for the Heart dataset.

|  | Gaussian | Epanechnikov | Exponential |
|---|---|---|---|
| Gaussian | $1.00 \pm 0.00$ | $0.16 \pm 0.04$ | $0.15 \pm 0.04$ |
| Epanechnikov | $0.16 \pm 0.04$ | $1.00 \pm 0.00$ | $0.50 \pm 0.06$ |
| Exponential | $0.15 \pm 0.04$ | $0.50 \pm 0.06$ | $1.00 \pm 0.00$ |

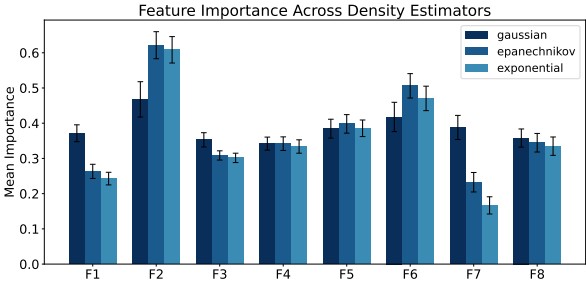

**Figure A.1.** The distribution of importance values according to the various types of kernels considered for the Diabetes dataset.

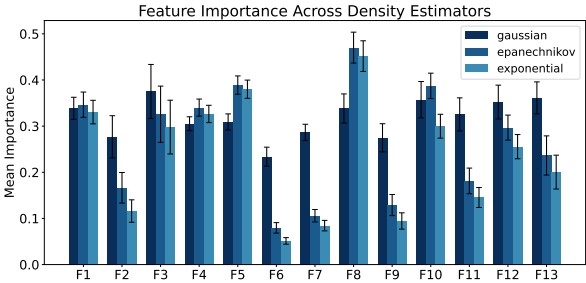

**Figure A.2.** The distribution of importance values according to the various types of kernels considered for the Heart dataset.

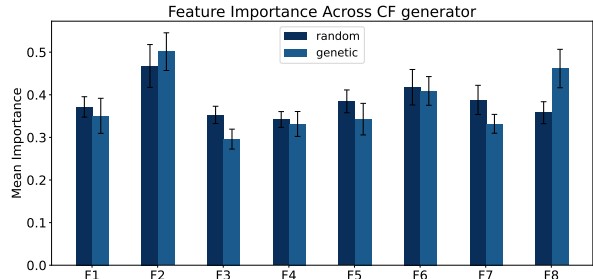

**Figure A.3.** The distribution of importance values according to the CF generation process considered for the Diabetes dataset.

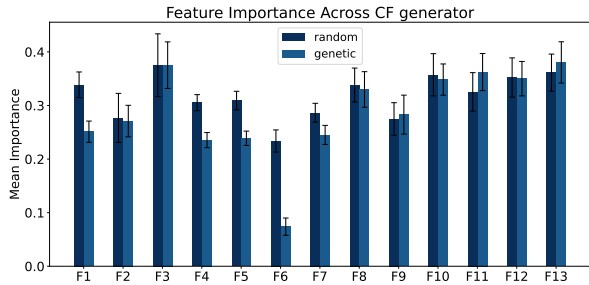

**Figure A.4.** The distribution of importance values according to the CF generation process considered for the Heart dataset.

**Table A.3.** Faithfulness results for the Diabetes dataset across various settings.

| Method | Comprehensiveness | Sufficiency |
|---|---|---|
| Gaussian | $\mathbf{0.5405 \pm 0.2559}$ | $-0.1288 \pm 0.1444$ |
| Epanechnikov | $0.1589 \pm 0.0278$ | $-0.0004 \pm 0.0147$ |
| Exponential | $0.1568 \pm 0.0276$ | $-0.0014 \pm 0.0149$ |
| Genetic | $0.1195 \pm 0.0247$ | $0.0506 \pm 0.0220$ |

**Table A.4.** Faithfulness results for the Heart dataset across various settings.

| Method | Comprehensiveness | Sufficiency |
|---|---|---|
| Gaussian | $\mathbf{1.8017 \pm 0.0965}$ | $2.2129 \pm 0.1357$ |
| Epanechnikov | $1.3794 \pm 0.0241$ | $2.1067 \pm 0.0167$ |
| Exponential | $1.3731 \pm 0.0236$ | $2.1127 \pm 0.0156$ |
| Genetic | $1.3007 \pm 0.0217$ | $2.1541 \pm 0.0208$ |

