# OpenReview forum: "CID: Measuring Feature Importance Through Counterfactual Distributions"
_NLDL.org/2026/Conference — NLDL 2026 Oral_

### Official Review · Reviewer_2ZQu · 2025-09-19
**Interesting proposed measure of feature importance**

**Rating:** 4
**Confidence:** 4
**Final Rating:** 4
**Final Confidence:** 4

**Summary:**

This work introduces a measure for feature importance called Counterfactual Importance Distribution (CID).
This measure relies on 1) generating a number of positive (changes the prediction) and negative (does not change the prediction) conterfactuals,
2) estimating the distributions of the positive and negative conterfactuals separately and 3) measuring the distance between the distributions
using a distance, $d_1$, inspired by the Jaccard distance.
They evaluate their method on two health datasets and compare agreement, comprehensiveness and sufficiency with three other methods: DiCE, SHAP and LIME.

**Strengths:**

**S1:** Understanding how ML models work is important, especially for high-stakes areas such as health applications.

**S2:** The proposed distance measure has a nice mathematical foundation.

**S3:** Relevant comparisons are made with other methods.

**Weaknesses:**

**Weaknesses:**

**W1:** Interpretation of the presented results is lacking. See **Q8**, **Q10** and **Q11**.


**Questions and Suggestions:**

**Q1:** Line 083: "Nevertheless in standard setting, We conclude the..." This sentence needs to be fixed.


**Q2:** Line 184-187:
"...to the best of our knowledge, our approach focuses on the novelty of measuring distributional differences stemming from counterfactuals."
I think you mean:
"...to the best of our knowledge, our approach of measuring distributional differences stemming from counterfactuals is novel."


**Q3:** Line 289-291: You define $d_k$ for any $k \geq 1$. In which situations would you use a $k \neq 1$?
If you can't think of any situations where you would use $k \neq 1$, then I suggest only defining $d_1$.


**Q4:** Proof in A1: $1-\delta\cdot(b-a) \leq \int_a^b max(p(x), q(x))dx \leq 1 + \delta\cdot(b-a)$,
so for $\int_a^c max(p(x), q(x))dx = \int_a^b max(p(x), q(x))dx + \int_b^c q(x)dx = \int_a^b max(p(x), q(x))dx + \epsilon$
we should have
$1-\delta\cdot(b-a) + \epsilon \leq \int_a^c max(p(x), q(x))dx \leq 1 + \delta\cdot(b-a) + \epsilon $.
So you are missing the $+\epsilon$ in the denominator in line 899.




**Q5:** Line 365: "$k \geq 2$", I think you mean "$k \geq 1$".


**Q6:** Section 4: In section 3 you define your distance and prove it is a metric. In section 4 you switch to talk about feature importance.
How exactly do you use your metric to calculate feature importance? Do you just say that if $d_1$ is large between distributions for
a specific feature for positive and negative counterfactuals, then feature importance is high?


**Q7:** Line 433-434: "reflecting that different values must be used for that variable to alter the model output." I don't understand this sentence.


**Q8:** Figure 1: It would be nice with some more explanation of how to interpret this figure.
E.g. Does the skin thickness plot mean that counterfactuals which changed the prediction mostly had very low skin thickness?


**Q9:** Line 449: Figure reference is missing.


**Q10:** Figure 2: Does this mean that using CID on the same model 100 times gives e.g. chol a feature importance varying between 0.1 and 0.9?
If yes, how do you interpret this high variability?


**Q11:** Figure 3: What does it mean that the predicted probablity is not monotonically decreasing?
Is there some way to explain that removing two features performs better than removing one?
Could it for example mean that the model uses a relation between features and not just the features themselves to predict the output?

**Final Justification:**

Following the rebuttal I still think that the work has a sound mathematical foundation and makes relevant comparisons with other methods in an important area of ML research. My recommendation is to accept.

**Justification:**

This work uses a mathematical foundation for their measure and make relevant comparisons with other methods in an important area of ML research. The work could be strengthened with more interpretation of their results, but all in all, it is solid work which is worth publishing.

---

> ### Author Rebuttal · Authors · 2025-10-16
>
> We thank the reviewer for carefully reading the paper. We provide detailed answers below to all questions and suggestions.
>
> # **Questions and Suggestions:**
>
> - **Q1+Q2+Q9**: Thank you for noticing this. We will fix the text for the updated version of the paper accordingly.
> - **Q3**: We thank the reviewer for this comment. As stated in line 306, our discussion focuses on the case $k = 1$, and all experiments are conducted using $d_1$. We agree that, in practice, it is most desirable to use $k = 1$. However, we chose to define $d_k$ for any $k \geq 1$ to emphasize the generality of our formulation. From a theoretical perspective, we believe it is valuable to show that $d_k$ satisfies the properties of a dissimilarity measure for any $k \geq 1$, thus highlighting the flexibility of our framework and allowing practitioners to adapt it to their specific needs.
> - **Q4**: Let us clarify the inequality in line 899. From the equation in line 897 we have that $d_k(p,q)=k- \frac{\int_a^b \min(p(x),q(x))dx}{\int_a^b \max(p(x),q(x))dx + \epsilon }$. Now, we know that $\min(p(x),q(x))<q(x)$ and so by integrating in $[a,b]$ we have that $\int_a^b \min(p(x),q(x))dx\leq \int_a^b q(x)=(1-\epsilon)$. On the other hand, $\max(p(x),q(x))\geq q(x)-\delta$ and so by integrating in $[a,b]$ we have that $\int_a^b \max(p(x),q(x))dx + \epsilon \geq \int_a^b q(x)-\delta\,dx +\epsilon = (1-\epsilon)-\delta(b-a)+\epsilon = 1-\delta(b-a)$. Combining the above inequalities we have $$\frac{\int_a^b \min(p(x),q(x))dx}{\int_a^b \max(p(x),q(x))dx + \epsilon } \leq \frac{1-\epsilon}{1-\delta(b-a)} $$
> and so by changing sign and adding $k$ we obtain the left inequality in line 899. Similarly, using inequalities in 907 you obtain the right inequality in line 899. Overall, the idea is that since $\int_a^b q(x) = 1-\epsilon$, the term $+\epsilon$ appearing in 897 cancels out.
> - **Q5**: Thank you for pointing this out. Indeed, $d_k$ is a dissimilarity measure for $k \geq 1$. In line 365, the condition $k \geq 2$ appears because this result follows directly from Proposition 3.2 (see Appendix A.3). However, the case $k \in [1, 2)$ requires a different and more elaborate mathematical proof (see Appendix A.4). For this reason, we decided to separate the two results to provide a clearer and more detailed explanation of each. In any case, as you correctly noted, $d_k$ is a dissimilarity measure for $k \geq 1$, and we will make sure to emphasize this in the text to avoid any confusion.
> - **Q6**: Yes, the computed value of distributional discrepancy directly represents the feature importance in our framework. The underlying intuition is as follows: if, for a given instance, the positive counterfactuals and the negative ones have similar distributions for a specific feature (i.e., a small distance), this feature does not significantly contribute to changing the model output and should therefore be assigned a low importance. Conversely for a large distance. Our design choice is to take the computed distance as the feature importance itself. As in other works, one could optionally define a threshold and assign binary importance [1],[2] (e.g., $1$ if the distance exceeds the threshold, $0$ otherwise), but such design choices are beyond the scope of this paper.
> - **Q7**:  In line 433-434 we meant that, in our context, if $d_1(p, q)$ is large for a given feature, the feature takes substantially different values in positive versus negative counterfactuals. In other words, changes in this feature are associated with changes in the model output. We will rewrite that sentence to clarify the message.
> - **Q8**: Thank you for pointing out the opportunity to include these details in the paper. You are correct that positive counterfactuals exhibit lower skin thickness values. Another insight is that, for the PedigreeFunction, smaller values of this feature are typically needed to influence the prediction. In contrast, for features such as Insulin and Glucose, the plot shows no clear preference for specific values to obtain positive or negative counterfactuals. Nevertheless, these observations are specific to this particular instance and cannot be generalized, but including them provides valuable context and enriches the interpretation of the plot.
> - **Q10**: The observed variability in feature importance across multiple runs of CID arises from several factors, including the complexity of the model, DiCE's use of random perturbations, and correlations between features in the dataset. While such variability exists, a possible solution would be to run the method multiple times and consider the average feature importance for each dimension, providing a more robust estimate. It is important to note that this example illustrates only one instance; overall, the more comprehensive faithfulness evaluations demonstrate that CID achieves better results. Nevertheless, variability in feature importance is a known issue and a recognized limitation of such methods, as discussed in [3–8].
> - **Q11:** The fact that the predicted probability is not monotonically decreasing can occur for several factors: the original instance may have values that reduce confidence and so by entering average values, the probability may rise above the original value; feature interactions or correlations: inserting a single feature may change the joint configuration in a way that increases confidence (for example, that feature may take values typical of the opposite class); non-linear models may exhibit non-additive effects. Importantly, Figure 3 is illustrative and refers to only one sample, thus this behavior may be specific to the instance.
>
> We thank the reviewer for their comments that will improve the interpretation of the results and overall quality of the paper.
>
> [1] Jin, H., Xue, A., You, W., Goel, S., & Wong, E. (2025). Probabilistic Stability Guarantees for Feature Attributions. *arXiv preprint arXiv:2504.13787*.
>
> [2] Ribeiro, M. T., Singh, S., & Guestrin, C. (2016, August). " Why should i trust you?" Explaining the predictions of any classifier. In *Proceedings of the 22nd ACM SIGKDD international conference on knowledge discovery and data mining* (pp. 1135-1144).
>
> [3] Krishna, S., Han, T., Gu, A., Wu, S., Jabbari, S., & Lakkaraju, H. (2022). The disagreement problem in explainable machine learning: A practitioner's perspective. *arXiv preprint arXiv:2202.01602*.
>
> [4] Goldwasser, J., & Hooker, G. (2024, January). Statistical Significance of Feature Importance Rankings. In *The 41st Conference on Uncertainty in Artificial Intelligence*.
>
> [5] Duan, J., Li, H., Zhang, H., Jiang, H., Xue, M., Sun, L., ... & Song, J. (2024, September). On the Evaluation Consistency of Attribution-Based Explanations. In *European Conference on Computer Vision* (pp. 206-224).
>
> [6] Ratul, Q. E. A., Serra, E., & Cuzzocrea, A. (2021, December). Evaluating attribution methods in machine learning interpretability. In *2021 IEEE International Conference on Big Data (Big Data)* (pp. 5239-5245). IEEE.
>
> [7] Alvarez-Melis, D., & Jaakkola, T. S. (2018). On the robustness of interpretability methods. *arXiv preprint arXiv:1806.08049*.
>
> [8] Harel, N., Obolski, U., & Gilad-Bachrach, R. (2022). Inherent inconsistencies of feature importance. *arXiv preprint arXiv:2206.08204*.

---

### Official Review · Reviewer_reje · 2025-10-03
**Review for "CID: Measuring Feature Importance Through Counterfactual Distributions"**

**Rating:** 4
**Confidence:** 3

**Summary:**

This work proposed a post-hoc local feature importance method called Counterfactual Importance Distri-
bution (CID). Positive and negative counterfactuals were leveraged to find the most important features. This was done by ranking the dimensions of the two types of counterfactuals and identifying where the two differed the most. To distinguish between the two counterfactuals, their distributions were measured using a new dissimilarity metric $d_1$, which utilised the degree of overlap between the two distributions. CID was evaluated on two datasets and compared to DiCE, LIME and SHAP to assess its effectiveness in terms of two suggested faithfulness metrics: sufficiency and comprehensiveness.  CID was highlighted to achieve improved faithfulness in some scenarios and offer complementary perspectives to existing approaches.

**Strengths:**

Overall, the work is well-motivated and aims to address some key gaps in local feature importance approaches. There was a good presentation of formulae and derivations. The dissimilarity metric and the faithfulness metrics presented fulfil the criteria for applications and evaluation of a diverse range of applications and offer new insights into model behaviours.

**Weaknesses:**

I found the discussion of the results lacking, why certain observations were important or what they could imply. For instance, the evaluation of faithfulness metrics in Tables 3 and 4 and the discussion of the Figure 3 results. I think the structure of the paper can be improved upon. For instance, in the text, Figure 2 is referenced before Figure 1; maybe they could be reorganised? Some points to consider: I) In 192, KDE is not defined when first referenced. II) Line 205, the word function appears twice, i.e. function function. III) Line 286, the word be might be missing. IV) Line 449 has issues with figure references, Figure ??. V) Something weird with lines 894 and 895 in A.1.V) Something to clarify the notation k being used in different contexts (top-k in feature alignment and k in dissimilarity measure), can one k be switched out for a different letter? VI) Clarify what L is in equation 4, perhaps. VII) The two xs in line 206 can be defined/ clarified when mentioned first.

**Justification:**

While showcasing a new, flexible local feature importance method which accounts for distributions of features, the work can be improved upon in terms of discussion and structuring of results. There are some issues that should be improved upon. But the proposed CID approach, along with the dissimilarity and fatihfulness metrics, is innovative and can provide new insights.

---

> ### Author Rebuttal · Authors · 2025-10-16
>
> We thank the reviewer for their useful insights that will improve the quality and exposition of the paper. We are going to address each weakness in detail.
>
> # Weaknesses
>
> - **W1**: Thank you for the suggestion. We will consider reorganizing the structure of the discussion in relation to Tables 3 and 4 and Figure 3. The main result of the tables is that CID shows significantly better results in terms of comprehensiveness, while in the Heart dataset, the results are still superior to the others but tied with those of LIME. In the case of sufficiency, the behavior is similar. Figure 3, on the other hand, is only intended to show the intuitive idea of what happens when we gradually remove the most important features for a particular instance, while the statistical results are summarized in the two tables. We will improve the clarity of the presentation of results in the updated version of the paper.
> - **W2**: You are indeed correct: we will make sure that Figure 1 is mentioned before than Figure 2 in Section 4.1.
>
> Note: Since there are two V points, we have reordered the points from V onwards, adding 1, in order to reference them correctly.
>
> - **Point I**: If you are referring to the acronym Kernel Density Estimation, we will ensure that the full term appears on line 192 with the abbreviation KDE. If, on the other hand, you are referring to which estimation method we are using, the idea is precisely not to specify the density estimation strategy, as the CID framework allows for a variety of choices, and we have adopted one of the possible options (Gaussian Kernels).
> - **Point II+III+IV+V+VIII:** Thank you for noticing these minor issues. We will make sure they are all corrected in the revised version.
> - **Point VI**: We acknowledge that both concepts use the variable $k$, but we believe the context clearly distinguishes their meanings. This choice was made to remain consistent with the common notation in XAI (top-k) and in parametric metrics. We will ensure that the final version of the paper makes this distinction explicit to avoid any possible confusion.
> - **Point VII**: As mentioned in line 508, we define $x=(x_1, \ldots, x_L)$, hence $L$ denotes the number of features, consistent with the notation used in the paper introducing these faithfulness metrics. We will make sure to clarify it in the updated version of the paper
>
> We appreciate the comments and we are going include corrections to further improve the clarity and presentation of the paper.

---

### Official Review · Reviewer_FdrB · 2025-10-09
**A well-structured paper introducing a counterfactual explanation method with promising results. I recommend acceptance.**

**Rating:** 4
**Confidence:** 3

**Summary:**

This paper introduces Counterfactual Importance Distribution (CID), a method that measures feature importance by comparing the distributions of successful and unsuccessful counterfactuals using an overlap-based metric.

Counterfactuals (CFs) explain a model's decision by showing what changes to the input would have changed the output.
Traditional CFs only find one or several examples that change the model's output.

The overlap-based metric quantifies how much two probability distributions share common support: if the distributions of positive and negative counterfactuals overlap heavily, the feature contributes little to changing the model's prediction. If they overlap less, the feature is more influential.

CID extends CFs by generating two sets of CFs: positive CFs ($C^+$) that successfully flip the model's prediction, and negative CFs ($C^-$) that fail to do so. It then models the distributions of these two sets using Kernel Density Estimation (KDE), and measures their difference via a newly proposed dissimilarity metric, $d_1(p,q)$, which is used to rank feature importance. Heree, $p$ and $q$ represent the kernel density estimates of the positive and negative CFs for a given feature, respectively. A larger $d_1$ value indicates greater feature importance.

Experiments on two benchmark datasets (Heart Disease and Diabetes) show that CID achieves higher faithfulness than other measures like SHAP, LIME, and DiCE, as indicated by improved comprehensiveness and sufficiency scores.

The paper is well written. Minor issues should be addressed, and some explanations could be added. Overall, I recommend acceptance.

**Strengths:**

- The authors propose a formulation of an overlap-based dissimilarity, defined in equations 1 and 2 (o(p,q) and d_k(p,q)). The metric is well-defined, and theoretically justified. The authors prove two results (Theorems 3.1 and 3.2) .

- The proposed method, CID, is compared against DiCE, SHAP, and LIME, which are three paradigms in local explainability. CID yields higher "comprehensiveness" and lower "sufficiency" on two standard datasets (Heart, and Diabetes)

- The example shows how overlap translates to a small dissimilarity when distributions mostly coincide.

- While further details could be provided for experiments, some configuration details are present for reproducibility.

- In general, the paper is well written and relatively easy to follow.

**Weaknesses:**

- Using probabilities (instead of logits) in equation $f(x)=P(y=M(x)\mid x)$ can create dataset-dependent scales. This might affect interpretability.

- Faithfulness metrics replace removed features with the global mean. In data with skew or class imbalance, this can over/under-estimate faithfulness.

- The authors refer to CID as "more stable and reliable" while also acknowledging "higher variance" on a dataset. This looks contradictory without a clear definition of stability.

**Justification:**

- In line 48, the authors mention that counterfactuals were "first introduced by [6]". However, counterfactual reasoning has been introduced long before by D. Lewis in 1973 [1]. Does [6] refer to the first use of CF in explainable AI ? If yes, the senetence should be clearer.

- In line 51, perhaps rewrite the sentence to clarify that counterfactual is about model outputs changing when input features are changed. In fact, in Machine Learning (ML), a counterfactual is more like: if X were different, what would the model's prediction be?
So instead of saying "If X had not occurred, Y would not have occurred", maybe you could say: "If input features X had taken different values, then model's output Y would have been different".

- In line 52, perhaps give an example of "why are they human-friendly explanations".

- In line 83, "We" is capitalized after a comma.

- In line 155, the paragraph jumps from Shapley Values to SHAP without clarifying that SHAP is an adaptation of Shapley values for ML.

- In line 159, perhaps define briefly "fairness" and "consistency".

- Perhaps precise that the comparison is done with local SHAP values (as mentionned for DiCE).

- Lines 165-167: "different feature importance methods can yield different rankings". Maybe cite some reasons of why they differ (different assumptions, metrics, or model behavior).

- In line 242, minor remark: maybe it would be better to change the notation of the variability "Var" to avoid misleading terminology, as "Var" is mostly used in the literature to denote the variance across a distribution.

- In Equation (1), put the comma a bit further so that it does not get mixed up with dx' .

- In line 317, $\mu$ is not defined


[1] https://perso.uclouvain.be/peter.verdee/counterfactuals/lewis.pdf

Questions:
- In what sense is CID "complementary" to existing methods? Does it identify features that others miss, or does it provide more reliable rankings?
- In line 136, what does "minimal changes to the input features" mean ? Does "minimal" mean fewest features changed ?
- Lines 183-187: how is the proposed approach different from the cited works [24–26] that also use CFs ?
- Reported experiments use binary classification. How can the proposed framework handle multi-class models ?
- How sensitive are results to KDE bandwidth ?
- How would marginal distributions capture correlations (as stated in Section 5). Isn't it that only joint distributions do that ?
- In Table 4, what is the negative sufficiency due to ?

---

> ### Author Rebuttal · Authors · 2025-10-16
>
> We thank the reviewer for carefully analyzing our paper. We will address weaknesses and questions in detail.
>
> # Weaknesses:
>
> - **W1+W2**: Thank you for raising these questions on how we evaluated faithfulness metrics. We decided to opt for simpler and standard choices in masking and evaluating faithfulness metrics [1-3] in order to maintain the focus of the paper on the feature importance strategy we propose, but you are right in that this option may lead to biased estimations, so we intend to highlight this in the updated version of the paper.
> On the other hand, we are not sure what you mean with dataset-depend scales; we would appreciate it if you could clarify it. We would like to stress that papers [39-41], mentioned in Section 4.3, employ the predicted probability as we do.
> - **W3**: You are indeed correct. The word “stable” in line 554 is used improperly; we meant faithful and we will correct this. Regarding the higher variance, this can be observed in the case of comprehensiveness for the Diabetes dataset, while in other cases there is no significant difference in variability. Although we highlight this aspect in Figure 2, it is only an illustrative example of a specific instance.
>
> # Justification:
>
> - **J1**: Yes, you are indeed correct. We will make sure to clarify this in the updated version.
> - **J2+J4+J5+J7+J9+J10**: Thank you for your suggestions and for pointing out these inaccuracies. We will resolve them in the updated version of the paper.
> - **J3**: An instance that can be added to the text for clarity can be the following from [4]: if a person named Peter is denied a loan, a counterfactual explanation could state: "If Peter earned 10,000 more per year, he would receive the loan." This type of explanation is easy to understand and directly relates changes in input features to changes in the predicted outcome, making it readily comprehensible.
> - **J6**: Thank you for pointing this out. Actually the three mathematical properties of SHAP are local accuracy, missingness and consistency ("fairness" was included by mistake). The three properties are well described in the paper [5] that we cite at the beginning of that paragraph and they would require to introduce some notations and explain several equations, thus we leave it to the reader to consult the reference.
> - **J8**: This is a critical aspect of feature importance methods: this stems from many sources as the data complexity, stochastic components of the method (such as Monte Carlo sampling in SHAP and LIME, or random perturbations in DiCE) or dataset properties. We will include in the paper some reference in this regard: [6-9].
> - **J11**: Thank you for noticing this, indeed $\mu$ is defined later in line 326, we will fix this.
>
> # Questions:
>
> - **Q1**: Table 1 and Table 2 highlights the low agreement between CID and the other 3 methods, which is computed using the feature agreement formula (eq. 3). The low values indicates that the features that CID detects as most relevant are different from the one for LIME, SHAP and DiCE, thus its complementary nature (i.e. suggests different dimensions of the data as most important).
> - **Q2**: The concept of minimal changes varies across the literature in counterfactuals and depends on the design of the optimization problem. For instance in [10] authors minimize a weighted version of the Manhattan distance to ensure that the distance between $x$ and the counterfactuals $x’$ is small. Alternatively, in [11] authors minimize the number of changed features and the Gower distance between $x$ and $x’$. In general, we require some metric $d$ to ensure that $x,x’$ are as similar as possible while ensuring a different prediction.
> - **Q3**: In [24], the authors integrate SHAP with counterfactual explanations. The idea is that, for a given instance, they evaluate how the predicted probability changes as features are progressively modified up to reaching a counterfactual instance. By doing so across multiple coalitions of features, they can attribute an importance score to each feature.
> [25] evaluates the importance of one feature by considering the frequency with which a slight change in the value of that specific feature results in a counterfactual instance (i.e., a significant change in the model’s outcome).
> Lastly, [26] authors combine DeepLIFT with counterfactuals to help choose among them. In particular they employ DeepLIFT as it compares the counterfactual examples to the initial instance and assigns the contribution scores according to the difference in predictions.
>
>     CID differs from the methods mentioned above: we consider distributional distances between two classes of counterfactuals with distinct outputs, thus not measuring point-wise changes, or employing other methods as SHAP or DeepLIFT on top of counterfactuals.
>
> - **Q5**: In the multi-class case with outputs $o_1, \ldots, o_s$, our framework can be naturally extended. Specifically, it is sufficient to consider any change in the output as a positive counterfactual, that is, if $x$ is such that $f(x) = o_i$, then a counterfactual $x'$ is positive when $f(x') = o_j$ with $j \neq i$, while the definition of negative counterfactuals remains unchanged. In this way, we still obtain two distributions of points, $C^+$ and $C^-$, and we can therefore apply the framework to estimate feature importance in the same manner as in the binary case. If domain knowledge is available, one might restrict the set of positive counterfactuals classes to those that are meaningful for the specific task and instance.
> - **Q6**: We thank the reviewer for this question. We have not investigated this direction. The framework we propose is flexible and allows for different choices (bandwidth, KDE, CF generator), and our approach employs Silverman's rule, a standard choice. In Appendix A.5, we show alternative CID settings for completeness, without intending to be exhaustive.
> - **Q7**: We agree that, strictly speaking, correlations between features are captured by joint distributions rather than marginal ones. However, our statement refers to the fact that, if the generation of $C^+$ and $C^-$ inherently considers the feature dependencies present in the data, then the resulting marginal distributions already reflect these dependencies indirectly. Although our metric $d_k$ operates per features, the samples used to estimate the distributions are drawn from counterfactual sets that encode feature relationships.
> - **Q8**: We thank the reviewer for this important point. A negative sufficiency value indicates that, when features are inserted in the order of estimated importance, the model's predicted probability can increase beyond the original instance's probability. This behavior can arise for several reasons. First, it may reflect feature interactions or correlations: inserting a single feature may change the joint configuration in a way that increases confidence (for example, that feature may take values typical of the opposite class, or it may interact with other features). Second, the original instance may have values that reduce confidence; by entering average values, the probability may rise above the original value. Lastly, non-linear models may exhibit non-additive effects.
>
> We appreciate the comments and we are going include corrections to further improve the quality of the paper.
>
> [1] I. C. Covert, S. Lundberg, and S.-I. Lee. “Explaining by removing: a unified framework for model explanation”. In: J. Mach. Learn. Res. (2021).
>
> [2] Sundararajan, M., Taly, A., & Yan, Q. (2017, July). Axiomatic attribution for deep networks. In *International conference on machine learning* (pp. 3319-3328). PMLR.
>
> [3] Hooker, S., Erhan, D., Kindermans, P. J., & Kim, B. (2019). A benchmark for interpretability methods in deep neural networks. *Advances in neural information processing systems*, *32*.
>
> [4] C. Molnar. Interpretable Machine Learning. A Guide for Making Black Box Models Explainable. 2nd ed. Independently published, 2022.
>
> [5] Lundberg, S. M., & Lee, S. I. (2017). A unified approach to interpreting model predictions. *Advances in neural information processing systems*, *30*.
>
> [6] Krishna, S., Han, T., Gu, A., Wu, S., Jabbari, S., & Lakkaraju, H. (2022). The disagreement problem in explainable machine learning: A practitioner's perspective. *arXiv preprint arXiv:2202.01602*.
>
> [7]  Duan, J., Li, H., Zhang, H., Jiang, H., Xue, M., Sun, L., ... & Song, J. (2024, September). On the Evaluation Consistency of Attribution-Based Explanations. In *European Conference on Computer Vision* (pp. 206-224).
>
> [8] Pawlicki, M. (2023, October). Towards quality measures for xAI algorithms: Explanation stability. In *2023 IEEE 10th International Conference on Data Science and Advanced Analytics (DSAA)* (pp. 1-10). IEEE.
>
> [9] Velmurugan, M., Ouyang, C., Moreira, C., & Sindhgatta, R. (2021, November). Evaluating stability of post-hoc explanations for business process predictions. In *International Conference on Service-Oriented Computing* (pp. 49-64). Cham: Springer International Publishing.
>
> [10] Wachter, S., Mittelstadt, B., & Russell, C. (2017). Counterfactual explanations without opening the black box: Automated decisions and the GDPR. *Harv. JL & Tech.*, *31*, 841.
>
> [11] S. Dandl, C. Molnar, M. Binder, and B. Bischl. “Multi-Objective Counterfactual Explanations”. In: Parallel Problem Solving from Nature – PPSN XVI. Springer International Publishing, 2020, pp. 448–469.

---

### Meta-Review · Area_Chair_BG3y · 2025-10-31

**Recommendation:** Accept (Oral)
**Confidence:** 4

**Metareview:**

The paper tackles an important and timely problem in explainable AI. The reviewers found the proposed CID metric mathematically well-founded and novel, with theoretical results (metric properties) and a reasonable practical validation on benchmark datasets. I would like to commend the authors on the writing and clarity of the paper and also the quality of their rebuttal.

While the empirical validation could be broader and there is work to do beyond what is currently in the paper, this work’s conceptual contribution and rigorous exposition make it a valuable addition to the literature on explainability.

---

### Decision · Program_Chairs · 2025-11-05

**Decision:**

Accept (Oral)

**Comment:**

We recommend an oral and a poster presentation given the AC and reviewers recommendations.